# Insights into Androgen Receptor Action in Lung Cancer

**Darko Durovski [1], Milica Jankovic [2] and Stefan Prekovic [3,\*]**

[1] Leiden University Medical Center, 2333 ZA Leiden, The Netherlands
[2] Department of Molecular Biology and Endocrinology, "Vinča" Institute of Nuclear Sciences,
   University of Belgrade, 11000 Belgrade, Serbia
[3] Center for Molecular Medicine, University Medical Center Utrecht, 3584 CG Utrecht, The Netherlands
[\*] Correspondence: s.prekovic@umcutrecht.nl; Tel.: +31-655576959

**Abstract:** Sex hormones and their receptors play a crucial role in human sexual dimorphism and have been traditionally associated with hormone-dependent cancers like breast, prostate, and endometrial cancer. However, recent research has broadened our understanding by revealing connections with other types of cancers, such as lung cancer, where the androgen receptor has been found to be particularly significant. This review aims to explore the molecular mechanisms of androgen action in lung cancer pathogenesis and progression, highlighting the potential of inhibiting the androgen receptor signaling pathway as a therapeutic strategy for lung cancer treatment.

**Keywords:** androgens; nuclear receptors; lung cancer

## 1. Introduction

Lung cancer stands as the foremost cause of cancer-related mortality globally, annually causing nearly 1.8 million fatalities [1], with an estimated 127,000 deaths projected for the year 2023 in the United States by the American Cancer Society [2]. This disease is often asymptomatic, diagnosed at a late stage [3], and characterized by the lowest 5-year relative survival rate out of all cancer types [4]. Importantly, reliable biomarkers for early stages of lung cancer are still missing [5]. Different types of lung cancer exist and can broadly be divided into two major groups: non-small cell lung cancer (NSCLC) and small cell lung cancer (SCLC). The former type, which includes lung squamous cell carcinoma, lung small cell carcinoma, and lung adenocarcinoma, accounts for 80% of all lung cancer cases [4]. The most common subtype of NSCLC is adenocarcinoma, which represents the most prevalent subtype among non-smokers [6]. SCLC is usually diagnosed in patients with smoking history and older than the age of 70, with a major fraction of these patients succumbing to the consequences of the disease rapidly [7].

The main risk factor associated with lung cancer development is tobacco smoke, which contains numerous carcinogens that can cause DNA damage [3]. Smoking-related DNA damage can lead to mutations in tumor suppressor genes, such as *p53*, which normally help prevent the accumulation of DNA damage and, thus, prevent the development of cancer. In addition to smoking, environmental and/or occupational exposure to pollution, chronic lung disease, lifestyle factors, and genetic factors may also contribute to the development of lung cancer [3]. Mutations in *EGFR*, *KRAS*, *BRAF*, and *HER2* have been identified as drivers of lung cancer [8], and the fusion of *ALK*, *ROS1*, and *RET* protein tyrosine kinase oncogenes with multiple partner genes has been linked to the development and progression of non-small cell lung cancer, particularly lung adenocarcinoma [9]. Recent studies have also suggested that sex hormone receptors and their corresponding ligands may play a role in the pathogenesis of lung cancer [10]. Sex hormone receptors, including androgen receptor (AR), estrogen receptors (ER) α and β, as well as progesterone receptor (PR)-A and PR-B, belong to the steroid receptor subgroup of the nuclear receptor (NR) superfamily, which consists of 48 members of transcription factors (TFs) with roles in metabolism, reproduction,

inflammation, and development [11]. Dysregulation of sex hormone signaling has been implicated in the development of different types of cancer, including prostate, breast, and lung cancer [10].

It has been reported that females have a higher incidence of lung cancer than males. However, despite frequent diagnosis at advanced stages, females tend to have better 5-year survival outcomes, and, in general, a better prognosis compared to the males [12]. Additionally, there are differences in subtype prevalence, with adenocarcinoma and bronchioloalveolar carcinoma being diagnosed more often in females than males, even after accounting for smoking status as a risk factor [13]. This points to a possible role of sex hormones and their receptors in the development and progression of lung cancer. As the role of ER in lung cancer biology has been extensively reviewed by others [8,14–16], the aim of this review is to provide a comprehensive overview of studies investigating the role of androgens and the AR in normal lung development and lung cancer. This includes information on association with survival and disease progression, molecular mechanisms, pathway crosstalk, and potential applications of inhibitors of AR signaling pathway. To that end, a literature analysis was conducted to collect information available concerning the link between the AR, its ligands, and lung cancer. Relevant literature was identified through search engines PubMed and Google Scholar, using the search terms (and their respective synonyms terms) "sex hormone", "androgen receptor (AR)", "androgen", "testosterone", "dihydrotestosterone", and "lung cancer" in different combinations. Only studies published in English were included in this review.

## 2. Androgens and the Androgen Receptor

Sex hormones are synthesized in the gonads (ovaries and testes), adrenal glands, or locally in other tissues [11]. Cholesterol serves as the precursor to all steroid hormones, including glucocorticoids and mineralocorticoids. Essential for the sex hormone signaling pathway, the enzyme P450-linked side chain cleaving enzyme, converts cholesterol into pregnenolone, which is subsequently used as a precursor for the synthesis of both estrogens and androgens. Pregnenolone is further converted to various, less potent androgens, which ultimately serve as backbones for testosterone biosynthesis. Most of the synthesized testosterone is bound to albumin and sex-binding hormone globulin (SBHG), and only 2–3% circulates free in the serum as biologically active [17,18]. In particular tissues, testosterone itself can be converted to a more potent dihydrotestosterone (DHT) hormone by the enzyme 5-$\alpha$ reductase [19]. Three isoenzymes of 5-$\alpha$ reductase exist: types 1, 2, and 3 [17]. Type 3 is of particular interest as it has been identified in both benign and malignant lung tissue and its overexpression has been detected in lung adenocarcinoma samples [20].

An androgen receptor (AR) is a member of the nuclear receptor (NR) family that shares structural similarities with other members, including estrogen receptors (ER), glucocorticoid receptors (GR), and progesterone receptors (PR). After androgens bind to AR, confirmation of the receptor changes and the protein translocates to the nucleus, where it binds to specific DNA response elements, known as hormone response elements (HREs), mostly found at enhancer regions, distal to AR target genes. After AR engages with HREs, nuclear proteins, such as co-activators and co-repressors, are recruited to ultimately influence transcriptional output of its target genes [6,21] (Figure 1). This is the classical mode of action of AR. Additionally, androgens can also induce rapid activation of other non-classical pathways in cells, including the MAPK pathway, PI3K/Akt pathway, and Src tyrosine kinase pathway [22].

The intricate structure of the AR is characterized by its multi-domain composition, consisting of three distinct functional units (Figure 1). These include an N-terminal transactivation domain (NTD), a DNA-binding domain (DBD), and a ligand-binding domain (LBD) [23]. The NTD houses an activation function (AF)-1 domain that works in tandem with co-regulatory proteins to modulate gene transcription. Meanwhile, the DBD is responsible for binding to specific DNA sequences, known as androgen response elements (AREs), which are strategically located in the promoter regions of target genes. Lastly, the LBD is

composed of two critical components, the AF-2 domain and the ligand-binding pocket. The AF-2 domain interacts with co-regulatory proteins, whereas the ligand-binding pocket binds to androgens. Notably, the hinge region connects the DBD and LBD and is vital for AR stability and function [23].

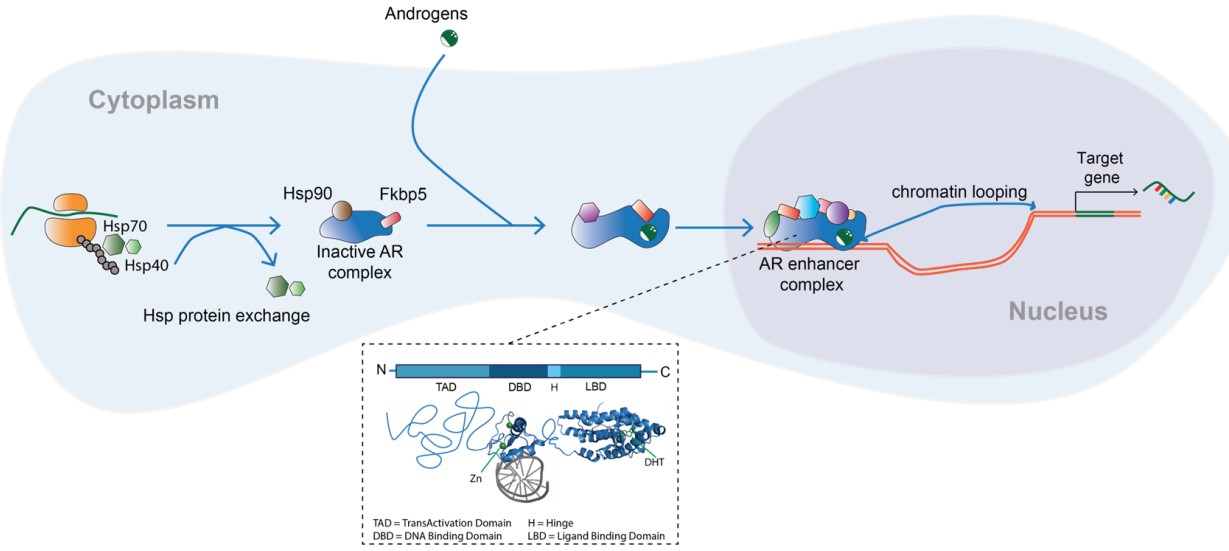

**Figure 1.** This scheme illustrates the intricate signaling mechanism of the androgen receptor and its corresponding gene/protein structure. Data from Protein Database (https://www.rcsb.org (accessed on 27 February 2023)); IDs 1R4I and 1T5Z.

The complexity of the AR is further illustrated by multiple splice variants, including the well-known AR-V7. The constitutively active AR-V7 lacks the ligand-binding domain (LBD) and has been linked to resistance to androgen deprivation therapy in prostate cancer patients [24]. Additionally, other splice variants, such as AR-V1, AR-V2, and AR-V3, could also play a role in the development and progression of prostate cancer [25]. Furthermore, mutations in AR can affect its activity and are associated with various diseases, such as androgen insensitivity syndrome (AIS) and prostate cancer [26]. These mutations can occur in different domains of the receptor, including the DNA-binding domain (DBD) and LBD. Mutations in the LBD can alter the specificity of ligand binding, enabling activation by other hormones, such as progesterone and estrogen [26].

The AR mediates effects of male sex hormones in reproductive and non-reproductive organ systems. Following its discovery in the 1960s, the expression of this receptor was demonstrated in several tissues, including bone, muscle, prostate, and adipose tissue [27]. Over the years, it became evident that the role of AR extends beyond sexual development, with its signaling contributing to various biological processes—for example, brain development, metabolism, and the immune system function [28]. Moreover, researchers demonstrated that AR plays a pivotal role in several pathologies, including androgen insensitivity syndrome, Kennedy's disease and cancer [29,30]. In particular, in terms of cancer, AR is the critical driver of prostate cancer; nonetheless, its dysregulation has also been proposed to contribute to the development of bladder, kidney, breast, and liver cancer [31,32].

## 3. Role of the Androgen Receptor in Lung Development

In addition to their role in sexual development, sex hormones play a role in the development of the lung. Mammalian lung development occurs in five consecutive histologically distinct stages: embryonic, pseudoglandular, canalicular, saccular, and alveolar [33,34]. Transitions between stages are under multifunctional molecular control, which involves growth factors, TFs, glucocorticoids, and sex hormones [34]. The fetal lung is exposed to

circulating sex hormones produced by C19 steroid precursors in maternal and fetal adrenals. Moreover, in fetal lungs, sex hormones can be inactivated by specific enzymes, adding another layer of regulation [33]. Normal lung development differs between males and females, with maturation and surfactant production occurring earlier in females compared to males [35,36].

Androgens produce distinct effects throughout lung development, depending on the stage. They can act as positive regulators in the process of branching morphogenesis in early development and as negative regulators responsible for delay in male lung maturation in late stages (Figure 2) [33]. Branching morphogenesis occurs at week five of gestation, with all airway branching occurring by week 24 of fetal life; AR is predominantly expressed in epithelial cells found at budding sites during this process [37]. The expression of AR was shown in both murine and human type II pneumocytes and bronchial epithelium [31]. The negative influence of androgens on the lung development was demonstrated in vivo, whereby male animals were found to have less developed lungs compared to their female counterparts [34,36]. Surfactant production in the fetal developing lung is under multi-factorial and multihormonal control. Glucocorticoids and estrogens can accelerate the production of surfactant phospholipids and proteins, while androgens can inhibit it [38]. The inhibitory role of testosterone and DHT on pulmonary surfactant production was shown in organ cultures and in vivo mouse and rabbit models, where the effects were attributed to hormone interactions with the AR found in lung fibroblasts and alveolar type II cells [39]. Despite the well-established role of AR in lung development, the precise role of AR in normal lung physiology remains uncertain and warrants further investigation.

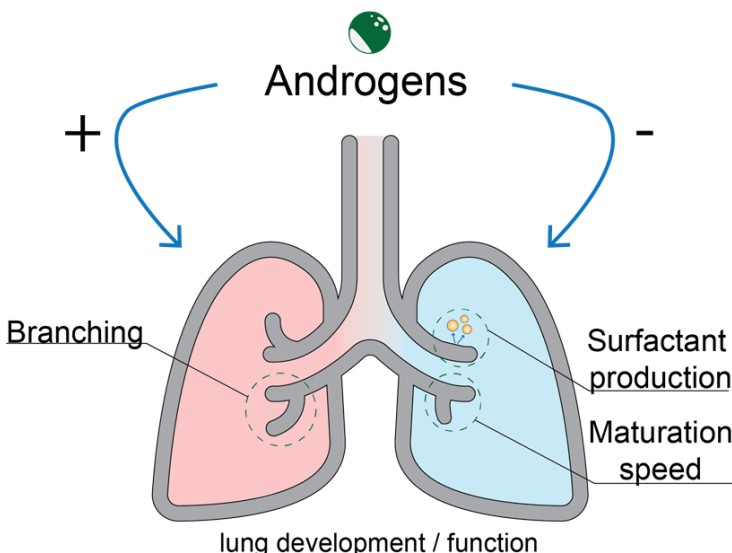

**Figure 2.** The scheme depicts the impact of androgens on lung development and function.

## 4. Androgens and the Androgen Receptor in Lung Cancer

### 4.1. Androgen Receptor Signaling in Lung Cancer

Early research efforts demonstrated a presence of the functional AR protein in lung cancer, specifically SCLC [40]. In addition to the presence of the receptor and its ability to readily bind the ligand, it was also shown that $5\alpha$-reductase is expressed in SCLC models, although at only 10% of the level found in prostate cancer cell lines. Since then, AR expression has also been demonstrated in the more prevalent subtype of lung cancer, NSCLC, both in cell lines and primary lung cancer material [19]. Following these observations, several studies have confirmed the presence of AR in cell line models of lung cancer [32,41]. Moreover, alterations of the AR (i.e., amplifications, deletions, or mutations) have been observed in 5% of lung squamous carcinoma and adenocarcinoma, further suggesting an important role in lung cancer [42].

In addition to the expression of the functional AR protein, activating enzymes that convert steroid precursors to biologically active sex hormones are expressed in lung cancer cell lines [33]. Provost et al. (2000) investigated the intrinsic capacity of A549 to metabolize sex hormones. Cells contained 17β-hydroxysteroid dehydrogenase (HSD) type 5, which catalyzes the transformation of 4-androstenedione into testosterone, and 3α-HSD type 3, which predominantly converts dehydroepiandrosterone (DHEA) to androstenediol, which can also activate the AR. Despite high expression levels of AR, low 3β-HSD activity was detected in an in vitro model, indicative of low capacity of the cell line to produce dihydrotestosterone (DHT) and testosterone themselves [39]. Furthermore, the expression of enzymes that inactivate sex hormones has also been demonstrated in lung cancer cells, as well as in mouse and human fetal lung tissue. For example, low hydroxysteroid 17-β dehydrogenase 6 (17β-HSD B6) expression has an impact on the progression of lung adenocarcinoma [43]. Furthermore, 17β-HSD B6 has both oxidoreductase and epimerase activities and an important function in androgen catabolism. It catalyzes the conversion of 3α-adiol to DHT and the conversion of androsterone to epi-androsterone, and its downregulation has already been linked to poor patient prognosis in hepatocellular carcinoma [44]. It has been demonstrated that 17β-HSD B6 is expressed at low levels in lung adenocarcinoma compared to normal lung tissue. Low expression of 17β-HSD B6 was associated with poor clinicopathological features, including tumor size, advanced tumor stage, low tumor differentiation, and poor patient prognosis [43]. Moreover, it was found that 17β-HSD B6 reduces radio resistance in lung cancer cells and suppresses proliferation, migration, invasion, and epithelial–mesenchymal transition. The research has found evidence that the potential mechanism includes activating PTEN expression and inhibiting AKT phosphorylation [45].

Androgens can affect proliferation of cancer cells. Testosterone is known to have a significant impact on protein synthesis rates, potentially resulting in heightened basal cell metabolism and increased energy [46]. Along the same line, AR activation in murine cells causes upregulation of genes involved in oxygen transport and heme biosynthesis and negative regulation of genes involved in apoptosis, DNA repair, and double-strand break repair [47]. The link between testosterone and hormone-dependent cancers (e.g., prostate cancer) has been well-researched, although with contrasting conclusions about its association to incidence [46].

Recently, studies have investigated the effect of androgens on cellular processes and signaling pathways in NSCLC. In AR-positive murine lung tissue and human NSCLC cell lines, androgen treatment induced profound changes to the transcriptional landscape. Particularly in the A549 cell line model, genes involved in oxygen transport and utilization were upregulated, while the ones involved in DNA repair and DNA recombination were downregulated, consistent with the proposed pro-carcinogenic role of androgens in lung cancer [47]. Wang et al. (2018) treated different NSCLC cell lines with agonists and antagonists of AR and found that the effect was dependent on the *KRAS* mutational status, proposing that a crosstalk between AR and KRAS exists [42]. This further suggests that AR co-operates with both *EGFR* (as well as other growth factor signaling pathways) and the Raf/MEK/ERK pathway linked to in various processes, such as differentiation, growth, chemotaxis, and apoptosis [48]. In relation to this, AR knockdown in NSCLC cell lines resulted in a decrease of cyclin D1 expression and subsequently led to inhibition of cell proliferation and anchorage-independent growth in vitro [49,50]. Additionally, suppression of AR expression was hallmarked by a decrease of OCT-4 protein expression, suggesting an interplay between the AR axis and the pathway affecting stemness and self-renewal [49].

Apart from the potential interaction between AR and KRAS, Recchia et al. (2009) shed light on the involvement of EGFR in DHT-induced growth stimulation of A549 lung cancer cells and LNCaP prostate tumor cells [6]. Their study uncovered that DHT treatment led to the up-regulation of CD1 and consequent cell proliferation, both of which could be mitigated by selective inhibitors or knockdown of either AR or EGFR. These findings

suggest that the cross-talk between AR and EGFR plays a significant role in the progression of prostate and lung cancer through the activation of the mTOR/CD1 pathway [6].

Compelling evidence for the role of AR in lung cancer comes from the study by Noronha et al. (1983). Female NZR-GD rats treated with a single dose of 20 mg/kg intraperitoneal administration of dimethylnitrosamine (DMN) typically develop lung, kidney, liver, and nasal cavity tumors [51]. It was observed that testosterone treatment before DMN injections resulted in enhanced lung tumor formation, while the same effects could not be observed in the cases of hepatocellular and kidney epithelial tumors [51]. In agreement with the latter are the findings coming from experiments performed in AR knockout (ARKO) mouse models [31]. The ARKO model was used in an NNK-BaP (eight doses of tobacco carcinogens, 4-(methylnitrosamino)-1-(3–pyridyl)-1-butanone (NNK) and benzo[a]pyrene (BaP))-induced mouse model of lung cancer; AR knockout resulted in reduced tumor size compared with control NNK-BaP mice [50].

In summary, the above-discussed research indicates that androgen receptor (AR) expression and activity have been observed in both SCLC and NSCLC. Alterations of the *AR* gene have been found in lung squamous carcinoma and adenocarcinoma. The expression of the enzymes that convert steroid precursors to biologically active sex hormones and those that inactivate them in lung cancer cells have been shown to impact lung adenocarcinoma progression. Androgens affect cancer cell proliferation and energy consumption, with AR activation leading to upregulation of genes involved in oxygen transport and downregulation of those involved in apoptosis and DNA repair (Figure 3). In vivo studies have revealed that DMN may enhance lung tumor formation, and that AR knockout results in reduced tumor size in mice (Figure 3). These findings suggest that targeting AR signaling may have potential therapeutic implications in lung cancer treatment.

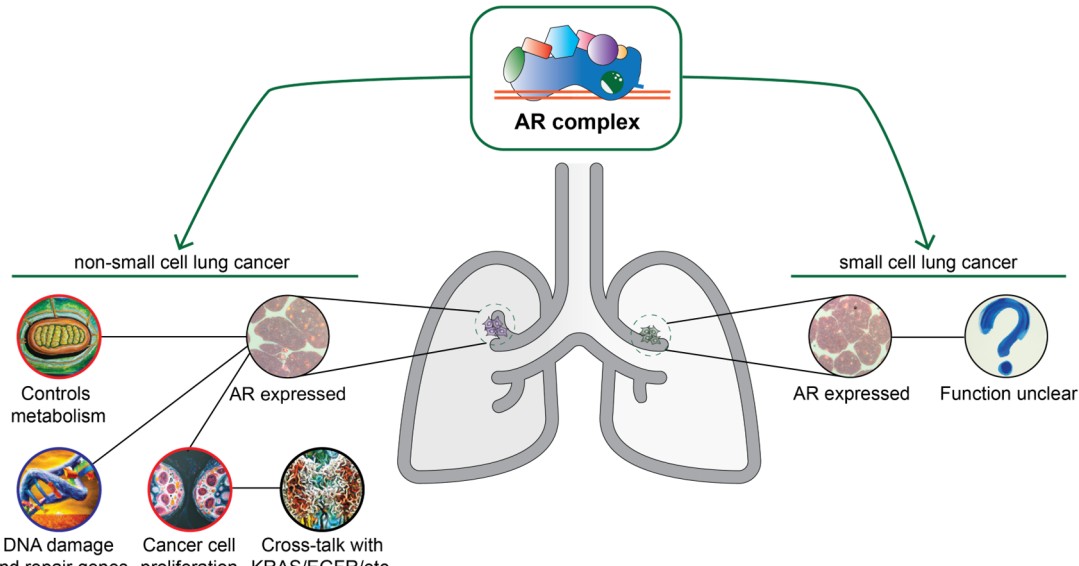

**Figure 3.** This scheme visually represents how active AR affects lung cancer, with red and blue outlines indicating positive and negative regulation, respectively. All the images were generated with the help of DALL·E 2, a tool developed by OpenAI (https://openai.com/product/dall-e-2 (accessed on 27 February 2023)).

### 4.2. Androgen Receptor as an Important Factor in Lung Cancer Biology: Evidence from Clinical Studies

The correlation between androgens, intra-tumor AR expression, and patient outcomes has been investigated in numerous studies. Using immunoassays, Hyde et al. (2012) investigated the relationship between total blood T levels and lung cancer risk in a cohort of 3635 community-dwelling elderly men. High T levels were associated with an increase

in lung cancer risk, even after accounting for smoking as a double confounder [18]. Using the samples of the same cohort, Chan et al. (2017) investigated the association between plasma T and DHT levels with lung cancer incidence by performing liquid chromatography–mass spectrometry (LC-MS). Comparable to the findings of the first study, higher T and DHT plasma levels were associated with an increased risk of developing lung cancer. One standard deviation (SD) increase in total T, equivalent to 4.87 nmol/L, significantly increased the adjusted risk of lung cancer (HR = 1.30, 95% CI 1.06–1.60; $p$ = 0.012). For every 1 SD (0.73 nmol/l) increase in DHT, the adjusted risk of lung cancer increased by 29% (HR = 1.29, 95% CI 1.08–1.54; $p$ = 0.004). When accounting for current smoking status, higher plasma levels of T did not have an association with lung cancer, however, DHT remained associated with increase in lung cancer risk (HR = 1.26, 95% CI 1.03–1.54; $p$ = 0.026) [41]. In contrast to these reports, Ørsted et al. (2014), who conducted a prospective cohort study (291 male and 193 female participants included, with a median follow up of 22 years), found no association between plasma T levels and the risk of developing lung cancer [46].

In respect to intra-tumor AR expression, Hattori et al. (2015) demonstrated that the AR is expressed only in a minor fraction (18/566 cases) of primary lung cancer tissues (which included squamous cell carcinoma, large neuroendocrine carcinoma, small cell carcinomas, and typical carcinoids) [52]. Irrespective of gender, an immunohistochemistry-based study by Grant et al. (2018) found no association between the AR expression levels and overall survival (OS) or recurrence of NSCLC in 136 tissue samples from Manitoba Tumor Bank (MTB) patients [12]. However, another study by Berardi et al. (2016) investigated the expression of sex hormone receptors in 62 patients with advanced NSCLC and found significant correlation with outcome. They linked the intra-tumor presence of AR, ER-α, and PgR to a significant correlation with outcomes in patients with advanced NSCLC. Patients with nuclear and cytoplasmic AR expression exhibited better survival when compared to those not expressing the protein (49 and 45 months, respectively, vs. 19 months)—although the number of patients demonstrating nuclear AR expression was relatively low (8/62) [53]. In contrast with the latter, Skjefstad et al. (2016), who collected tumor tissues from 335 patients and evaluated the expression of AR in the cytoplasm and nucleus of tumor epithelial and stromal cells, found that both male and female patients had worse outcomes with respect to disease-specific survival compared to the rest of the cohort [54].

Examination of mutation profiles in cell-free DNA found in plasma samples of lung cancer patients revealed that 48.28% of patients exhibited the androgen receptor p.H875Y mutation, which had previously been detected in prostate and breast cancer [55]. Moreover, the researchers found that cancer samples with AR mutation showed a higher total mutational burden compared to the samples without it. Previous reports have shown that mutations of residue 875 of AR expand the ligand repertoire of the receptor, allowing for activation by other steroid hormones [26]. However, the exact functional relevance of this mutation in lung carcinogenesis is yet to be investigated [55].

Furthermore, Rades et al. (2012) investigated the association of AR, ER-α, and PR expression with locoregional control, metastases-free survival, and OS in stage II/III NSCLC tumors of patients who received radiotherapy (RT) [4]. Of importance, while ER-α expression was associated with worse treatment outcomes with regards to locoregional control, metastasis-free survival, and OS in both females and males, no significant association was found between AR or PR expression and these parameters [4]. Regarding OS, a trend towards improved locoregional control was seen for lower N classification and not smoking during RT [4].

The relationship between androgens and lung cancer risk has been studied extensively, with some studies showing that high levels of total testosterone and dihydrotestosterone (DHT) are associated with increased risk of developing lung cancer. However, other studies have found no association between plasma testosterone levels and lung cancer risk. The expression of AR within primary lung cancer tissues is limited, but some studies have shown a correlation between the presence of AR, ER-α, and PgR and better outcomes in

patients with late-stage metastatic non-small cell lung cancer (NSCLC). In contrast, other studies have found that patients with AR expression in tumor epithelial and stromal cells had worse outcomes in terms of disease-specific survival. A mutation of the *AR* gene, p.H875Y, has been found in plasma samples of lung cancer patients and may play a role in lung carcinogenesis, but its functional relevance is yet to be fully understood. In patients with stage II/III NSCLC who received radiotherapy, no association was found between AR expression and treatment outcome, while ER-$\alpha$ expression was associated with worse treatment outcomes.

### 4.3. Targeting of the Androgen Receptor Pathway

Blocking the androgen receptor signaling pathway can be achieved in several ways, specifically by either targeting AR or the enzymes responsible for the conversion of steroid precursors into T or DHT (as reviewed in detail in [56]). At the level of biosynthesis, CYP17A1 and 5-$\alpha$ reductase inhibitors, such as abiraterone and finasteride, respectively, may be used to block the enzymes responsible for production of T, and ultimately its conversion to DHT. Non-steroidal antiandrogens, such as bicalutamide, nilutamide, flutamide, or the more recently developed enzalutamide, can bind to AR and engage in competitive binding with testosterone and DHT [57]. Steroidal antiandrogens, such as cyproterone acetate, competitively inhibit the action of androgens and the secretion of luteinizing hormone (LH), which leads to decreased testosterone production. Lastly, gonadotropin-releasing hormone (GnRH) agonists, which include buserelin, goserelin, and leuprolide can lead to a downregulation of GnRH receptors and a subsequent decrease in LH and testosterone. [13].

Several studies have investigated the effects of pharmacologically targeting the androgen receptor pathway in lung cancer. In NSLC cell lines, inhibition of AR by enzalutamide has been suggested to cause radiosensitization [42]. Furthermore, expression of miR-224-5p, an exosome-secreted microRNA targeting AR, has been shown in NSCLC cancer specimens. More importantly, lentivirus-mediated inhibition of the expression of miR-224-5p led to decreased proliferative and migratory capacity of A549 and H1299 cells. Using flow cytometry, it was shown that overexpression of miR-224-5p accelerated progression through the cell cycle and inhibited apoptosis. Finally, using a tumor xenograft mouse model, researchers found that injection of A549 cells overexpressing miR-224-5p led to an increase in growth and appearance of necrotic regions in the lungs, along with higher expression of the epithelial-to-mesenchymal transition (EMT) marker N-cadherin [58].

Harlos et al. (2015) investigated the effect of androgen pathway manipulation (APM) on survival in 3018 men diagnosed with lung cancer. Of the total patient population, 339 had used a form of APM, either in the form of an antiandrogen, 5-alpha reductase inhibitor, or GnRH [13]. Patients who received APM, either partial or complete, after their lung cancer diagnosis (HR = 0.36, $p = 0.0007$) and those who were exposed to APM both before and after their diagnosis (HR = 0.53, $p < 0.0001$) had a significantly better survival compared to the patients who did not use any APM [13]. Following stratification of the patient sample based on disease stage, researchers found that exposure to APM was significantly associated with longer survival in early-stage disease if treatment occurred after diagnosis. In late-stage disease, a difference was significant in those exposed to APM after diagnosis (HR = 0.41, $p = 0.02$) and before and after diagnosis (HR = 0.54, $p < 0.0001$). The effect appeared to diminish in patients who were diagnosed with late-stage lung cancer and with each passing year of receiving therapy after diagnosis [13].

Recently, a proteasome-based system with proteolysis-targeting chimeras (PROTACs) targeting AR has been developed and its effects examined in lung cancer cell lines [1]. PROTACs are heterobifunctional small molecules that harness the ubiquitin-proteasome system to degrade the target protein. It was demonstrated that one of the enzalutamide-based PROTACs with a specific exit vector led to a dose-dependent reduction of viability in the A549 lung cancer cell line model [59].

In light of recent studies, it has become apparent that the androgen receptor and its signaling axis may hold promise as a therapeutic target for NSCLC. Pharmacological

targeting of the androgen pathway has shown potential in NSCLC cell lines, with clinical research also demonstrating a correlation between androgen pathway manipulation and improved survival outcomes in lung cancer patients. These findings offer a compelling case for further investigation and development of androgen receptor-targeted therapies for NSCLC treatment.

## 5. Discussion

A major challenge in understanding the role of androgens in lung cancer is the inconsistent reporting of smoking status in clinical studies. Smoking can confound the effect of androgens on lung cancer as it both increases levels of unbound testosterone in healthy men and independently increases the incidence of lung cancer [18,60]. Furthermore, women who smoke have a higher risk of developing lung cancer compared to non-smoking women [16,61]. To complicate matters further, crosstalk between nuclear receptors exists, and studies on menopausal status and the risk of developing lung cancer have been inconclusive [62,63]. Given these confounding factors, future studies should aim to separate cohorts based on smoking and menopausal status to better understand the effect of androgens on lung cancer.

In addition to these challenges, the role of androgens in SCLC is still not well understood. This is likely due to the lower prevalence of this subtype and the fact that most clinical studies have focused on early rather than late-stage NSCLC, which can have a significant influence on survival analysis.

The collection time of blood or other biological specimens is crucial in studying testosterone levels, as testosterone exhibits diurnal variation with peaks and troughs throughout the day in males [64]. This diurnal variation can confound the results of epidemiological studies if the collection time of samples is not standardized. For instance, samples collected in the morning will likely have higher testosterone levels than those collected in the evening [65]. Therefore, it is important to collect samples at the same time of day to minimize the potential impact of diurnal variation.

Another challenge in assessing the effect of androgens on lung cancer is the measurement of free testosterone levels. Testosterone is mostly found circulating in its bound form to albumin or SBHG, and less than 2% is found in its free, bioactive form [18]. Blood-based association studies should, thus, include measurements of free testosterone, and LC-MS/MS should be used for more precise quantification compared to immunoassays [41]. Additionally, the localization of AR expression in stained tumor tissues may have an influence on the results, yet many immunohistochemistry studies do not specify this localization [53].

While research on the role of androgens in lung cancer has provided valuable insights, there are still many challenges that need to be addressed. Future studies should aim to better account for confounding factors, such as smoking and menopausal status, and use more precise methods for measuring free testosterone levels. These efforts will enable a more accurate understanding of the role of androgens in lung cancer and improve the development of targeted therapies for patients.

## 6. Conclusions

Given the growing body of research on the role of androgens and the AR in lung cancer, there is increasing interest in exploring their potential as therapeutic targets. Pharmacologically modulating the AR signaling pathway may hold promise as a novel approach to treating patients with lung cancer. To achieve this, a deeper understanding of the exact mechanisms underlying AR action in lung cancer, both at the pathway and genomic levels, is needed. This could involve probing and analyzing high-throughput datasets, such as those on mutation burden and gene expression, to identify new biomarkers and potential therapeutic targets for lung cancer. Additionally, investigations into post-translational modifications of AR could offer insights into the biochemical regulation and cellular function of this protein, providing new avenues for research and potential therapeutic interventions. Overall, further research in this area could have significant clinical implications, potentially

leading to the development of novel diagnostic and treatment options for patients with this deadly disease.

**Author Contributions:** Investigation: D.D., M.J. and S.P.; writing—review and editing: D.D., M.J. and S.P. All authors have read and agreed to the published version of the manuscript.

**Funding:** This work was supported by the Alpe d'Huzes/KWF Bas Mulder Award. M.J. was supported by the Ministry for Science and Technological Development of the Republic of Serbia (451-03-47/2023-01/200017).

**Institutional Review Board Statement:** Not applicable.

**Informed Consent Statement:** Not applicable.

**Data Availability Statement:** Not applicable.

**Acknowledgments:** We express our gratitude to our colleagues for their valuable contributions and insightful feedback.

**Conflicts of Interest:** The authors declare no conflict of interest.

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
