# Peer review of "Insights into Androgen Receptor Action in Lung Cancer"

_endocrines, doi:10.3390/endocrines4020022_

Round 1

Reviewer 1 Report

Durovski et al review literature on androgen receptor action in lung cancer. They highlight the studies elucidating the molecular mechanisms behind androgen action and how this contributes to pathogenesis and progression of lung cancer. The authors do a nice job on underscoring the potential of AR inhibition as a therapeutic strategy for treatment of lung cancer. The manuscript is well written and encompasses a breadth of topics to nicely highlight AR function in lung development and lung cancer. The manuscript will be elevated and be more impactful with a few additional components.

Please consider addressing the following points.

Major Comments:

1)    Important mention of AR mutations in patient samples. Is there any expression of AR splice variants in different disease stages? If so, please add to the text.

2)    Does AR expression alter with known lung cancer genetic factors – EGFR, KRAS, BRAF, or HER2? Could the authors please expand on this point. Studies have shown the potential crosstalk of AR and KRAS, it will be helpful to highlight if there are genetic alterations among AR and these factors in patient tissue.

3)    The manuscript will benefit from a schematic depicting the function and expression location of AR in lung development highlighting the unique roles in males and females as discussed in the text for section 3.

4)    Additionally, the manuscript will benefit from a schematic visually showcasing the role of AR in different disease stages – NSCLC and SCLC as discussed in section 4.1. This will be a useful image to be utilized by readers in the field to summarize the studies discussed by the authors in text.

Minor Comments:

1)    Please use the most recent reference for updated statistics of lung cancer prevalence – i.e., Siegel et al 2023.

2)    Add a summary statement at the end of section 4.2 on line 200 or 211 to conclude the findings from the in vitro and in vivo studies mentioned to indicate to the reader the take home message of AR function in lung cancer.

3)    Also add a summary statement at the end of section 4.3 on line 310 to highlight the potential to target AR and its signaling axis in lung cancer therapy.

4)    Discussion point – consider adding note on collection time of blood, specimens, etc…since testosterone has peaks and values throughout the day in males. Please discuss how this could confound results for epi studies.

Author Response

  • Durovski et al review literature on androgen receptor action in lung cancer. They highlight the studies elucidating the molecular mechanisms behind androgen action and how this contributes to pathogenesis and progression of lung cancer. The authors do a nice job on underscoring the potential of AR inhibition as a therapeutic strategy for treatment of lung cancer. The manuscript is well written and encompasses a breadth of topics to nicely highlight AR function in lung development and lung cancer. The manuscript will be elevated and be more impactful with a few additional components.

We would like to thank the reviewer for the time invested and all the constructive comments.

  • Please consider addressing the following points.

Major Comments:

  • Important mention of AR mutations in patient samples. Is there any expression of AR splice variants in different disease stages? If so, please add to the text.

We thank the reviewer for this comment. In relation to this we have now expanded the section on AR biology, indicating how AR splice variants may contribute to disease initiation and progression. Unfortunately, in terms of lung cancer the importance of splice variants of AR has not been studied. The new paragraph highlighting this has been added:

“The complexity of the AR is further illustrated by multiple splice variants, including the well-known AR-V7. The constitutively active AR-V7 lacks the ligand-binding domain (LBD) and has been linked to resistance to androgen deprivation therapy in prostate cancer patients [25].Additionally, other splice variants like AR-V1, AR-V2, and AR-V3 could also play a role in the development and progression of prostate cancer [26]. Fur-thermore, mutations in AR can affect its activity and are associated with various diseases, such as androgen insensitivity syndrome (AIS) and prostate cancer [27]. These mutations can occur in different domains of the receptor, including the DNA-binding domain (DBD) and LBD. Mutations in the LBD can alter the specificity of ligand binding, enabling ac-tivation by other hormones, like progesterone and estrogen [27]..”

  • Does AR expression alter with known lung cancer genetic factors – EGFR, KRAS, BRAF, or HER2? Could the authors please expand on this point. Studies have shown the potential crosstalk of AR and KRAS, it will be helpful to highlight if there are genetic alterations among AR and these factors in patient tissue.

We thank the reviewer for this comment, in addition to discussion we have now included a paragraph:

“Apart from the potential interaction between AR and KRAS, Recchia et al. (2009) shed light on the involvement of EGFR in DHT-induced growth stimulation of A549 lung cancer cells and LNCaP prostate tumor cells [52]. Their study uncovered that DHT treatment led to the up-regulation of CD1 and consequent cell proliferation, both of which could be mitigated by selective inhibitors or knockdown of either AR or EGFR. These findings suggest that the cross-talk between AR and EGFR plays a significant role in the progression of prostate and lung cancer through the activation of the mTOR/CD1 pathway [52].

  • The manuscript will benefit from a schematic depicting the function and expression location of AR in lung development highlighting the unique roles in males and females as discussed in the text for section 3.

We have now included a figure (Figure 2) depicting the abovementioned section.

  • Additionally, the manuscript will benefit from a schematic visually showcasing the role of AR in different disease stages – NSCLC and SCLC as discussed in section 4.1. This will be a useful image to be utilized by readers in the field to summarize the studies discussed by the authors in text.

We have now included a figure (Figure 3) depicting the abovementioned section.

Minor Comments:

  • Please use the most recent reference for updated statistics of lung cancer prevalence – i.e., Siegel et al

We have now included this reference in the opening sentence of the manuscript:

“Lung cancer stands as the foremost cause of cancer-related mortality globally, annually causing nearly 1.8 million fatalities [1], with an estimated 127,000 deaths projected for the year 2023 in the United States by the American Cancer Society.”

  • Add a summary statement at the end of section 4.2 on line 200 or 211 to conclude the findings from the in vitroand in vivo studies mentioned to indicate to the reader the take home message of AR function in lung cancer.

In line with reviewers comment we have added the following:

“The relationship between androgens and lung cancer risk has been studied extensively, with some studies showing that high levels of total testosterone and dihydrotestosterone (DHT) are associated with increased risk of developing lung cancer. However, other studies have found no association between plasma testosterone levels and lung cancer risk. The expression of AR within primary lung cancer tissues is limited, but some studies have shown a correlation between the presence of AR, ER-α, and PgR and better outcomes in patients with late-stage metastatic non-small cell lung cancer (NSCLC). In contrast, other studies have found that patients with AR expression in tumor epithelial and stromal cells had worse outcomes in terms of disease-specific survival. A mutation of the AR gene, p.H875Y, has been found in plasma samples of lung cancer patients and may play a role in lung carcinogenesis, but its functional relevance is yet to be fully understood. In pa-tients with stage II/III NSCLC who received radiotherapy, no association was found between AR expression and treatment outcome, while ER-α expression was associated with worse treatment outcomes.

  • Also add a summary statement at the end of section 4.3 on line 310 to highlight the potential to target AR and its signaling axis in lung cancer therapy.

In line with reviewers comment we have added the following:

“In light of recent studies, it has become apparent that the androgen receptor and its signaling axis may hold promise as a therapeutic target for NSCLC. Pharmacological targeting of the androgen pathway has shown potential in NSCLC cell lines, with clinical research also demonstrating a correlation between androgen pathway manipulation and improved survival outcomes in lung cancer patients. These findings offer a compelling case for further investigation and development of androgen receptor-targeted therapies for NSCLC treatment..”

  • Discussion point – consider adding note on collection time of blood, specimens, etc…since testosterone has peaks and values throughout the day in males. Please discuss how this could confound results for epi studies.

We thank the reviewer for this comment. We have now added a new paragraph to the discussion:

“The collection time of blood or other biological specimens is crucial in studying testos-terone levels, as testosterone exhibits diurnal variation with peaks and troughs throughout the day in males [66]. This diurnal variation can confound the results of epidemiological studies if the collection time of samples is not standardized. For instance, samples collected in the morning will likely have higher testosterone levels than those collected in the evening [67]. Therefore, it is important to collect samples at the same time of day to minimize the potential impact of diurnal variation.”

Reviewer 2 Report

This review article is intended to evaluate the influence of androgen receptors on lung cancer.

Gene names must be in italics (lines 36, 191, 193).

The introduction does not include also gene fusion targets for lung cancer (for example ALK, ROS1). Also, although it talks about cancer in smokers and non-smokers, the p53 gene is not mentioned.

It would be very nice if the detailed flowchart of the literature review process would be reported in a picture or scheme.

Line 55 probably should be a written review instead manuscript.

A diagram of the action of androgens should be provided, and what is shown in the diagram should not be the same described in the text.

Chapters 3 and 4 should be condensed and concretized there is some redundant information.

Author Response

  • This review article is intended to evaluate the influence of androgen receptors on lung cancer

We would like thank the reviewer for the time and constructive comments that helped us improve the manuscript.

  • Gene names must be in italics (lines 36, 191, 193).

All gene names have been checked and changed to italics.

  • The introduction does not include also gene fusion targets for lung cancer (for example ALK, ROS1). Also, although it talks about cancer in smokers and non-smokers, the p53 gene is not mentioned.

We would like to thank the reviewer for this comment, we have now included this in the introduction:

“The main risk factor associated with lung cancer development is tobacco smoke, which contains numerous carcinogens that can cause DNA damage [3]. Smoking-related DNA damage can lead to mutations in tumor suppressor genes such as p53, which normally help prevent the accumulation of DNA damage and thus prevent the development of cancer.”

  • It would be very nice if the detailed flowchart of the literature review process would be reported in a picture or scheme.

We thank the reviewer for this comment, however as we have now added new figures (Figures 1-3) and worked a lot on improving the text/readability, we have unfortunately not incorporated this comment, and hope that the reviewer understands our reasoning.

  • Line 55 probably should be a written review instead manuscript.

This has now been corrected.

  • A diagram of the action of androgens should be provided, and what is shown in the diagram should not be the same described in the text.

We have now added a figure representing the androgen receptor signaling axis – Figure 1 in the new version of the manuscript.

  • Chapters 3 and 4 should be condensed and concretized there is some redundant information.

We have worked on making the whole manuscript more readable, which also includes Chapters 3 and 4.

Reviewer 3 Report

In this manuscript the authors depict the role of androgen receptor in lung cancer.

The manuscript is clear and simple to read but needs some modifications.

In Par.3, lanes 114-122, the description of the alveoli seems to be inappropriate in this context and stops the lung development description. For the same reason, the sentence at lanes 123-124 should be added after the description of the role of AR in lung development and modified.

In Par4.1, lanes 154-176, all the data about the enzymes should be contextualized and linked to AR (considering that the topic of this manuscript is the role of AR in lung cancer).

Lane 177, the authors write:”…an increase in testosterone levels…”. Is this increase in circulation testosterone levels or in local (lung tissue) testosterone levels? Please specify.

The authors should re-organize the Discussion. They could ideally divide this paragraph in two and summarize, on one side, all the data about NSCLC and, on the other, all the data about SCLC.

Author Response

  • In this manuscript the authors depict the role of androgen receptor in lung cancer. 

We would like thank the reviewer for the time and constructive comments that helped us improve the manuscript.

  • The manuscript is clear and simple to read but needs some modifications.

We have improved the manuscript and readability in line with comments from all the reviewers.

  • In Par.3, lanes 114-122, the description of the alveoli seems to be inappropriate in this context and stops the lung development description. For the same reason, the sentence at lanes 123-124 should be added after the description of the role of AR in lung development and modified.

This has now been removed/amended in the new version of the manuscript.

  • In Par4.1, lanes 154-176, all the data about the enzymes should be contextualized and linked to AR (considering that the topic of this manuscript is the role of AR in lung cancer).

We have now linked this to AR function.

“In addition to the expression of the functional AR protein, activating enzymes that convert steroid precursors to biologically active sex hormones are expressed in lung cancer cell lines [34]. Provost et al. (2000) investigated the intrinsic capacity of A549 to metabolize sex hormones. Cells contained 17β-hydroxysteroid dehydrogenase (HSD) type 5, which catalyzes the transformation of 4-androstenedione into testosterone and 3α-HSD type 3 which predominantly converts dehydroepiandrosterone (DHEA) to androstenediol, which can also activate the AR. Despite high expression levels of AR, low 3β-HSD activity was detected in an in vitro model indicative of low capacity of the cell line to produce dihydrotestosterone (DHT) and testosterone themselves [40]. Furthermore, the expression of enzymes that inactivate sex hormones has also been demonstrated in lung cancer cells, as well as in mouse and human fetal lung tissue. For example, low hydroxysteroid 17-β dehydrogenase 6 (17β-HSD B6) expression has an impact on the progression of lung adenocarcinoma[44]. 17β-HSD B6 has both oxidoreductase and epimerase activities and an important function in androgen catabolism. It catalyzes the conversion of 3α-adiol to DHT and the conversion of androsterone to epi-androsterone, and its downregulation has al-ready been linked to poor patient prognosis in hepatocellular carcinoma [45]. It has been demonstrated that 17β-HSD B6 is expressed at low levels in lung adenocarcinoma compared to normal lung tissue. Low expression of 17β-HSD B6 was associated with poor clinicopathological features, including tumor size, advanced tumor stage, low tumor differentiation, and poor patient prognosis [44]. Moreover, it was found that 17β-HSD B6 reduces radioresistance in lung cancer cells and that it suppresses proliferation, migration, invasion, and epithelial-mesenchymal transition. The research has found evidence that the potential mechanism includes activating PTEN expression and inhibiting AKT phosphorylation [46].

  • Lane 177, the authors write:”…an increase in testosterone levels…”. Is this increase in circulation testosterone levels or in local (lung tissue) testosterone levels? Please specify.

Thank you so much for this comment. We have now rephrased this:

“Androgens can affect proliferation of cancer cells. Testosterone is known to have a sig-nificant impact on protein synthesis rates, potentially resulting in heightened basal cell metabolism and increased energy [47].”

  • The authors should re-organize the Discussion. They could ideally divide this paragraph in two and summarize, on one side, all the data about NSCLC and, on the other, all the data about SCLC.

The discussion section has now been reorganized in relation to all the reviewer’s comments.

“A major challenge in understanding the role of androgens in lung cancer is the inconsistent reporting of smoking status in clinical studies. Smoking can confound the effect of androgens on lung cancer as it both increases levels of unbound testosterone in healthy men and independently increases the incidence of lung cancer [19, 62]. Furthermore, women who smoke have a higher risk of developing lung cancer compared to non-smoking women [17, 63]. To complicate matters further, crosstalk between nuclear receptors exists, and studies on menopausal status and the risk of developing lung cancer are inconclusive [64, 65]. Given these confounding factors, future studies should aim to separate cohorts based on smoking and menopausal status to better understand the effect of androgens on lung cancer.

In addition to these challenges, the role of androgens in SCLC is still not well understood. This is likely due to the lower prevalence of this subtype and the fact that most clinical studies have focused on early rather than late-stage NSCLC, which can have a significant influence on survival analysis.

The collection time of blood or other biological specimens is crucial in studying testosterone levels, as testosterone exhibits diurnal variation with peaks and troughs throughout the day in males [66]. This diurnal variation can confound the results of epidemiological studies if the collection time of samples is not standardized. For instance, samples collected in the morning will likely have higher testosterone levels than those collected in the evening [67]. Therefore, it is important to collect samples at the same time of day to minimize the potential impact of diurnal variation.

Another challenge in assessing the effect of androgens on lung cancer is the measurement of free testosterone levels. Testosterone is mostly found circulating in its bound form to albumin or SBHG, and less than 2% is found in its free, bioactive form [19]. Blood-based association studies should thus include measurements of free testosterone, and LC-MS/MS should be used for more precise quantification compared to immunoassays [42]. Additionally, the localization of AR expression in stained tumor tissues may have an influence on the results, yet many immunohistochemistry studies do not specify this localization [55].

While research on the role of androgens in lung cancer has provided valuable insights, there are still many challenges that need to be addressed. Future studies should aim to better account for confounding factors such as smoking and menopausal status, and use more precise methods for measuring free testosterone levels. These efforts will enable a more accurate understanding of the role of androgens in lung cancer and improve the development of targeted therapies for patients.”

Reviewer 4 Report

Durovski et al. present a well-rounded review of the potential role of Androgen Receptor (AR) in lung cancer. The roles of sex hormones and associated receptors are often linked to the primary tissues that natively express these molecules, but their biochemical and cellular roles are often poorly explored in other tissue types. AR is a classic example of a nuclear receptor that is understudied in this context.

The manuscript sheds key insights on the prognostic significance of AR, its mode of signaling, expression and localization. But the authors primarily rely on the few studies that relate to the role of AR to lung development and cancer onset/progression. The authors are strongly advised to probe and include highthroughput datasets which have been deposited to web-based repositories and are accessible through customized and user-friendly online tools. Information on mutation burden (COSMIC/ c-Bioportal), normal tissue gene expression (GTEX), tumor vs normal gene expression (TCGA, GEPIA) and finally prognosis (KM plotter, ProgGene) would all be very useful to the readers to understand and guage the significance of a gene that is poorly characterized in lung and other tissues. Appropriate figures/ visualizations that capture all this information should be added to bolster the manuscript. Additionally, the authors are strongly encouraged to include a figure depicting the canonical signaling pathway(s) linked to AR. Finally, post-translational modifications on AR have been studied to a significant extent and are intricately linked to the biochemical regulation and hence cellular function of the protein. A section on this is vital and must be included in the manuscript. PhosphositePlus is a valuable database that has catalogued such PTMs on specific residues and their biochemical significance. A domain structure figure of AR that highlights/displays some of the biochemically essential PTMs would be good to include. The authors have produced a good review of the protein and including relevant information supported by figures will enable broader interest.

The authors are advised to carefully proof-read their manuscript as some minor typo/language errors persist.

Author Response

Durovski et al. present a well-rounded review of the potential role of Androgen Receptor (AR) in lung cancer. The roles of sex hormones and associated receptors are often linked to the primary tissues that natively express these molecules, but their biochemical and cellular roles are often poorly explored in other tissue types. AR is a classic example of a nuclear receptor that is understudied in this context.

We would like to thank the reviewer for praising our review.

  • The manuscript sheds key insights on the prognostic significance of AR, its mode of signaling, expression and localization. But the authors primarily rely on the few studies that relate to the role of AR to lung development and cancer onset/progression. The authors are strongly advised to probe and include highthroughput datasets which have been deposited to web-based repositories and are accessible through customized and user-friendly online tools. Information on mutation burden (COSMIC/ c-Bioportal), normal tissue gene expression (GTEX), tumor vs normal gene expression (TCGA, GEPIA) and finally prognosis (KM plotter, ProgGene) would all be very useful to the readers to understand and guage the significance of a gene that is poorly characterized in lung and other tissues. Appropriate figures/ visualizations that capture all this information should be added to bolster the manuscript. Additionally, the authors are strongly encouraged to include a figure depicting the canonical signaling pathway(s) linked to AR. Finally, post-translational modifications on AR have been studied to a significant extent and are intricately linked to the biochemical regulation and hence cellular function of the protein. A section on this is vital and must be included in the manuscript. PhosphositePlus is a valuable database that has catalogued such PTMs on specific residues and their biochemical significance. A domain structure figure of AR that highlights/displays some of the biochemically essential PTMs would be good to include. The authors have produced a good review of the protein and including relevant information supported by figures will enable broader interest.

We absolutely agree that AR is understudied in terms of its molecular biology in lung cancer, however in light of our manuscript being a review paper we have not included any of the re-analysis of public data. We hope that the reviewer understands that even if a lot of these datasets are easily accessible, in order for the analysis to actually bring something new they would need to re-analyzed and validated in other datasets. We have now added a statement in the conclusion section of the manuscript addressing the need to study AR using various publicly available data-sets as good bases for hypothesis generation and further studies.

“Given the growing body of research on the role of androgens and the AR in lung cancer, there is increasing interest in exploring their potential as therapeutic targets. Pharma-cologically modulating the AR signaling pathway may hold promise as a novel approach to treating patients with lung cancer. To achieve this, a deeper understanding of the exact mechanisms underlying AR action in lung cancer, both at the pathway and genomic levels, is needed. This could involve probing and analyzing high-throughput datasets, such as those on mutation burden and gene expression, to identify new biomarkers and potential therapeutic targets for lung cancer. Additionally, investigations into post-translational modifications of AR could offer insights into the biochemical regulation and cellular function of this protein, providing new avenues for research and potential therapeutic interventions. Overall, further research in this area could have significant clinical implications, potentially leading to the development of novel diagnostic and treatment options for patients with this deadly disease.”

  • The authors are advised to carefully proof-read their manuscript as some minor typo/language errors persist.

We have proof-read and corrected all the minor typo/language errors found.

Round 2

Reviewer 1 Report

The authors have adequately addressed all comments in a thorough manner.

Reviewer 2 Report

Authors should be congratulated for their work. They satisfactorily answered all my concerns. The manuscript is suitable for publication.

Reviewer 3 Report

The authors exhaustively reply to all my comments.

Reviewer 4 Report

Comments were addressed.